# Community-based complex interventions to sustain independence in older people, stratified by frailty: a protocol for a systematic review and network meta-analysis

Thomas F Crocker ![ORCID],[1] Andrew Clegg ![ORCID],[1] Richard D. Riley,[2] Natalie Lam,[1] Ram Bajpai ![ORCID],[1,3] Magda Jordão,[1] Eleftheria Patetsini,[1] Ridha Ramiz,[1] Joie Ensor,[2] Anne Forster,[1] John R F Gladman[4,5,6,7]

► Prepublication history and additional materials for this paper is available online. To view these files, please visit the journal online (http://dx.doi.org/10.1136/bmjopen-2020-045637).

For numbered affiliations see end of article.

**Correspondence to**
Dr Thomas F Crocker;
tom.crocker@bthft.nhs.uk

## ABSTRACT

**Introduction** Maintaining independence is a primary goal of community health and care services for older people, but there is currently insufficient guidance about which services to implement. Therefore, we aim to synthesise evidence on the effectiveness of community-based complex interventions to sustain independence for older people, including the effect of frailty, and group interventions to identify the best configurations.

**Methods and analysis** Systematic review and network meta-analysis (NMA). We will include randomised controlled trials (RCTs) and cluster RCTs of community-based complex interventions to sustain independence for older people living at home (mean age ≥65 years), compared with usual care or another complex intervention. We will search MEDLINE (1946 to September 2020), Embase (1947 to September 2020), CINAHL (1981 to September 2020), PsycINFO (1806 to September 2020), CENTRAL and clinical trial registries from inception to September 2020, without date/language restrictions, and scan included papers' reference lists. Main outcomes were: living at home, activities of daily living (basic/instrumental), home-care services usage, hospitalisation, care home admission, costs and cost effectiveness. Additional outcomes were: health status, depression, loneliness, falls and mortality. Interventions will be coded, summarised and grouped. An NMA using a multivariate random-effects model for each outcome separately will determine the relative effects of different complex interventions. For each outcome, we will produce summary effect estimates for each pair of treatments in the network, with 95% CI, ranking plots and measures, and the borrowing of strength statistic. Inconsistency will be examined using a 'design-by-treatment interaction' model. We will assess risk of bias (Cochrane tool V.2) and certainty of evidence using the Grading of Recommendations Assessment, Development and Evaluation for NMA approach.

**Ethics and dissemination** This research will use aggregated, anonymised, published data. Findings will be reported according to Preferred Reporting Items for Systematic Reviews and Meta-Analyses guidance. They

## Strengths and limitations of this study

► This will be the first systematic review with network meta-analysis (NMA) comparing the effectiveness of community-based complex interventions to sustain independence for older people, including the effect of frailty and pre-frailty.

► A careful process to group interventions, including summarising each intervention with the Template for Intervention Description and Replication, will produce an analysis that is transparent and relevant to policy-makers, commissioners and providers.

► In addition to the direct treatment effects, indirect treatment effects will be analysed using a random-effects NMA allowing us to compare different service models with each other.

► Summary of findings tables developed using the Grading of Recommendations Assessment, Development and Evaluation approach for NMA will provide an accessible assessment of the certainty and size of treatment effects.

► The review is likely to be limited by lack of detail about the experimental conditions and wider care system in some trials, and the lack of consistent outcome measures.

will be disseminated to policy-makers, commissioners and providers, and via conferences and scientific journals.
**PROSPERO registration number** CRD42019162195.

## INTRODUCTION

Global population projections indicate that older people are the fastest growing demographic, with the percentage of people aged 65 years and over expected to almost double by 2050,[1] and similar projections for developed countries such as the UK.[2] Given this success of increasing lifespan, current policy and initiatives such as the WHO's Decade of Healthy Ageing emphasise increasing the

number of years lived in good health.[3] [4] This focus on sustaining health is crucial for enabling people to realise their strong preference for living with independence within a community.[5] Additionally, older people are core users of health and care services, so the ageing population demographic has profound implications for service planning and delivery. However, there is currently insufficient guidance for policy-makers, commissioners and providers about which community services should be implemented.

Frailty is an especially problematic feature of population ageing, with increased risk of losing independence, hospitalisation, care home admission and mortality.[6] In the UK, around 10% of people aged 65 years and over have frailty, rising to around 50% of people aged over 85 years.[7] UK NHS expenditure increases considerably with advancing age, with a threefold increase for people aged over 70 years.[8] UK social care expenditure for older people is expected to rise to £12.7 billion by 2022.[9] Extra annual cost to the healthcare system per person was £561.05 for mild, £1208.60 for moderate and £2108.20 for severe frailty with reference to 2013/2014 UK costs.[10] This estimates a total additional cost of £5.8 billion per year across the UK.[10] These findings are mirrored in other developed countries.

There is a critical evidence gap regarding which community-based interventions are clinically and cost effective for older people, including those living with frailty and pre-frailty. This evidence gap means that there is considerable uncertainty regarding the appropriateness of interventions and how they should best be configured and commissioned. Previous systematic reviews and meta-analyses (MAs) have reported evidence for clinical and cost effectiveness of community-based complex interventions for reducing hospital admission, nursing home admission, falls and functional decline in older people.[11–13] However, previous reviews have not used network meta-analysis (NMA) to summarise whether different types of interventions have differential effects on outcomes, limiting usefulness for policymakers, health and social care commissioners and providers. Moreover, few systematic reviews have investigated such interventions delivered outside of the home environment. A landmark 2008 systematic review and MA summarised evidence from 89 trials, including 97 984 people.[11] The review reported that, in general, complex interventions provided in the community are effective for older people but lacked detail about what types of complex care improve outcomes, and does not include studies published over the last decade, which are potentially influential. This review only considered frailty in relation to one intervention (comprehensive geriatric assessment) and used a disability-based, non-validated definition of frailty to categorise trials. Standard MA techniques were used to synthesise the evidence.

Recognising that research evidence, understanding of frailty and MA methods have advanced considerably in the last decade, the review requires a contemporary update to identify how interventions might best be configured

to improve outcomes and inform commissioning and delivery of evidence-based services.

Specific review questions are:

1. Do community-based complex interventions to sustain independence in older people increase living at home, independence and health-related quality of life?
2. Do community-based complex interventions to sustain independence in older people reduce home-care requirement, depression, loneliness, falls, hospitalisation, care home admission, costs and mortality?
3. How should interventions be grouped for network meta-analysis (NMA)?
4. What is the optimal configuration of community-based complex interventions to sustain independence in older people?
5. Do intervention effects differ by frailty level (not frail, pre-frailty; frailty)?

## Objectives

The overall aim of this systematic review is to synthesise evidence on the effectiveness of community-based complex interventions to sustain independence in older people, including the effect of frailty and pre-frailty, and group interventions to identify the best configurations. For this systematic review, we define sustaining independence to mean maintaining or improving independence in activities of daily living (washing, dressing, grooming, toileting, walking, preparing meals, doing housework, managing finances, assisting others, etc), but not only one of these specific activities (eg, walking only). The specific objectives are as follows:

1. To identify randomised controlled trials (RCTs) and cluster RCTs (cRCTs) of community-based complex interventions to sustain independence in older people.
2. To synthesise evidence of their effectiveness for key outcomes in an MA of study-level data.
3. To identify key intervention components and study-level frailty to inform groupings for NMA and meta-regression.
4. To compare effectiveness of different intervention configurations using NMA.
5. To investigate the impact of frailty and pre-frailty using meta-regression.

## METHODS AND ANALYSIS

This protocol is reported in accordance with the reporting guidance provided in the Preferred Reporting Items for Systematic Reviews and Meta-Analyses Protocols (PRISMA-P) statement and PRISMA-NMA reporting guidelines.[14] [15]

## Eligibility criteria

We will select studies according to their design and the PICO criteria: participants, intervention, comparator and outcome(s) of interest.

## Study design

RCTs and cRCTs are eligible. Where only one unit of randomisation (an individual or cluster) is allocated to an arm of a trial, we will exclude the trial as the treatment effect is completely confounded with the unit. We accept minimisation as a method of sequence generation, in keeping with the Cochrane risk of bias (RoB) guidance. Crossover trials are also eligible; however, we will only use outcome data from the pre-crossover period.

## Participants (population)

Older people living at home (mean age of participants: 65 years or older). We will exclude trials of residents of care/nursing homes as these are the subject of other large-scale reviews.[16 17] If not all participants are living at home, we will only include the trial if data can be extracted specifically for these participants.

## Intervention

Aligned with our focus on community-based complex interventions, trials will be considered eligible if:

► The intervention is both initiated and mainly provided in the community.
► The intervention includes two or more interacting components (intervention practices, structural elements and contextual factors).
► The intervention is targeted at the individual person, with provision of appropriate specialist care.
► A focus of the intervention is sustaining (maintaining or improving) the person's independence.

A broad range of interventions will potentially be eligible, which may differ in terms of how the service is organised and what is done to or for the older person. Interventions may meet our criteria for including two or more interacting components by including multiple discrete practices, such as exercise sessions and nutritional advice. Other eligible interventions could include one practice that interacts with other structural elements such as being reliant on general practice or other services; or interaction with contextual factors by being substantially tailored to the person's physical and social environment. Examples would include comprehensive geriatric assessment or rehabilitation interventions.

Interventions that would not be considered eligible for inclusion are as follows:

► The intervention is either not initiated, or not mainly provided, in the community, or neither. For example, interventions delivered in outpatient, day hospital, inpatient and intermediate (post-acute) care settings.
► The intervention includes only one discrete component (intervention practices, structural elements and contextual factors) such as a drug, treadmill training, yoga, provision of information, cataract surgery, hearing aid, medication review and nutritional supplements.
► The intervention is not targeted at the individual person, with provision of appropriate specialist care, for example, general staff education (not training in

a patient-level intervention), practice-level reorganisation, operational, managerial or IT interventions, public health messages.
► A focus of the intervention is not sustaining (maintaining or improving) independence in activities of daily living (ADL). For example, interventions that primarily address cognitive deficits, mood disorders, or both will be excluded, unless they also aimed to improve overall independence.
► Condition-specific interventions, for example, case management for older people with diabetes, chronic obstructive pulmonary disease or depression.
► Interventions in which the primary focus is falls prevention as this evidence is already well synthesised, including in a recent NMA.[18] Nonetheless, falls will be a key additional outcome in this review.

## Comparator

Usual care, 'placebo' or attention control, or a different complex intervention meeting our criteria are eligible comparators.

## Outcome(s)

Studies will be included where outcome data were recorded at a minimum 24-week timepoint. For all outcomes of interest, data will be extracted and categorised for three timepoints: around 6 months, around 12 months and around 24 months.

### Main outcomes

► The main outcomes are living at home (defined either as a reported trial outcome, or the inverse of care home admission and mortality if reported separately); independence in ADL (basic/instrumental); home-care services (non-healthcare professional) usage; hospitalisation; care home admission; costs; and cost effectiveness.

### Additional outcomes

► The additional outcomes are health status/health-related quality of life, depression, loneliness, falls and mortality.

This update to the landmark 2008 systematic review and MA by Beswick and colleagues[11] refines the criteria used by that review, which will lead to the exclusion of some of their included studies; we recount these differences here. We will exclude falls prevention studies as a recent NMA has been conducted in that area.[18] Our criteria exclude interventions that are initiated in hospital and those conducted in outpatient settings, to ensure the interventions are firmly placed in the community. We will also exclude interventions in residential care settings, as these are already the subject of large-scale reviews, and the different settings provide different opportunities and challenges for intervening. Finally, we will exclude studies without an intervention targeted at the older person, for example, financial incentives for general practitioners, for consistency within our NMA.

## Search strategy

Search strategies have been developed and tested through an iterative process by an experienced medical information specialist in consultation with the review team. We will search the following databases from inception:

► Cochrane Central Register of Controlled Trials (CENTRAL; issue 9 of 12, September 2020).
► MEDLINE Ovid (1946–September 2020).
► Embase and Embase Classic Ovid (1947–September 2020).
► CINAHL EBSCO (1981–September 2020).
► PsycINFO Ovid (1806–September 2020).

We will also search trial registers (ClinicalTrials.gov and the International Clinical Trials Registry) from inception and scan reference lists of included papers. Publication status, date or language restrictions will not be used, and translation will be arranged as necessary throughout the process. A draft search strategy for MEDLINE is provided in the online supplemental appendix A. A PRISMA flow chart will be presented showing the process of study selection (online supplemental figure S1).[14]

## Study selection

Following deduplication, search results will be imported into the Rayyan web application (https://rayyan.qcri.org/). Two researchers will independently assess the title and abstract of each record. We will obtain full text articles for all potentially eligible trials. Study selection will be conducted by two researchers with guidance from the project management group (PMG), and disagreements will be resolved by consensus discussion involving the PMG. We will contact study authors if further information is required.

## Data collection process

Two researchers will independently extract data using a piloted data extraction form in a purpose-built Microsoft Access database. Characteristics of included and excluded studies' tables will be produced in Review Manager (RevMan) V.5.4. Summary of findings tables will be produced in Grading of Recommendations Assessment, Development and Evaluation (GRADE) Pro.[19]

## Intervention grouping

We will group interventions for NMA in a three-stage process.
1. We will use the Template for Intervention Description and Replication (TIDieR) framework to summarise reported interventions (including comparators).[20] The TIDieR framework includes 12 key items, including the why, what, who provided, how, where, when and how much of the intervention, including the broader healthcare context.
2. We will complete a content analysis of the summarised interventions using the TIDieR framework in nVivo V.12 to inform provisional groupings.[21]
3. We will develop provisional intervention groupings based on the service organisation or structure (eg,

team structure), key patient care processes (eg, assessment and follow-up) and specific patient care interventions (eg, exercise, ADL practice and relaxation). The intervention types will become the nodes in the NMA.

## Assessment of frailty

We anticipate that a range of validated instruments and operationalised measures will be used to identify pre-frailty and frailty in included trial populations of some studies. Examples of such frailty measures include: the use of the Fried phenotype model, the Tilburg Frailty Indicator, Groningen Frailty Indicator, Study of Osteoporotic Fractures criteria, Chinese Canadian study of health and ageing clinical frailty scale, Hebrew Rehabilitation Center for Aged Vulnerability Index, Vulnerable Elders Survey, and Brief frailty measure derived from the Canadian study of health and ageing or a formally produced Frailty Index.[22] We will classify the trial population in accordance with the frailty measure, so long as it is developed or validated according to the modern meaning of frailty and not as a generic term for being old or disabled. We will report methods used for each trial, including cut-off points for identification of pre-frailty and frailty.

We also anticipate that many studies will not formally have described study populations in terms of frailty. In such circumstances, two reviewers with extensive clinical academic frailty expertise (AC and JRFG) will independently use the well-validated phenotype model as a framework to categorise study-level frailty profile (not frail, pre-frailty and frailty) of trial participants if the relevant variables are reported.[23] The model is based on five characteristics (weight loss; exhaustion; low energy expenditure; slow gait speed and low grip strength). Evidence of ≥3 indicates frailty, 1–2 pre-frailty and 0 not frail. In the remaining studies where neither a recognised frailty measure nor the variables needed to apply the frailty phenotype categorisation are reported, the two reviewers will independently attempt to classify the populations based on trial eligibility criteria and/or reported baseline characteristics closely linked to frailty, including gait speed, hand grip strength, mobility, activity or disability levels. Any disagreements will be resolved by consensus.

In categorising study-level frailty, we recognise that trials may include participants across different frailty categories, so as well as 'not frail', 'pre-frail' and 'frail'; our categories will also include 'not frail and pre-frail', 'pre-frail and frail' and 'all'.

Our main analysis of the impact of frailty will only include trials that used a validated measure. Trials in which the reviewers allocated a study-level frailty level on the basis of eligibility criteria and/or baseline characteristics will be examined in secondary analyses.

## Intention to treat and missing data

If both per-protocol and intention-to-treat analyses are reported for a trial, we will prioritise intention-to-treat data.[24] In all instances, we will report whether analysis was conducted on data that were complete, complete

after imputation or incomplete, and we will examine and report any material differences in results across these types. When results for main outcomes are missing for a trial, we will contact authors to request the missing data.

## RoB within individual studies

Researchers will independently assess the RoB of each result of interest from each included trial, using Cochrane's RoB 2—a revised tool for assessing RoB in randomised trials.[24] For cRCTs, we will additionally assess identification/recruitment bias, and the other issues such as loss of clusters, detailed in version 6 of the Cochrane Handbook.[25] For each domain in the RoB 2 tool, a judgement of high RoB, low RoB or some concerns will be made, then an overall risk-of-bias judgement will be reached for each assessed outcome, with any disagreements resolved by consensus.

## Summary measures

For each trial and each outcome separately, effect estimates and confidence intervals (CIs) will be extracted comparing intervention and control groups. For continuous outcomes, we aim to extract the intervention effect as mean differences. We will consider using standardised mean difference if different measures are used for similar constructs. For binary outcomes, we will calculate risk ratios (RRs) and odds ratios (ORs). For survival (time-to-event) outcomes, hazard (rate) ratios (HRs) will be extracted. Any details about non-proportional hazards will also be extracted. Outcomes at all timepoints will be recorded and grouped appropriately (around 6 months, around 12 months and around 24 months). Where effect estimates and/or CIs are not available, we will use other information (eg, p values, means for each group at follow-up, etc) to derive the information indirectly.

## Unit of analysis issues

We will apply adjustment for trials that use cluster randomisation without adjusting standard errors.[25] As intraclass correlations needed to make such correction are rarely reported, we will use values obtained from external literature for the outcome examined (or if these are not available, use a single plausible value and examine the impact of varying this value in sensitivity analysis).

## Examination of potential effect modifiers

Treatment effect modifiers relate to methodological or clinical characteristics of the trials that influence the magnitude of treatment effects (on a given scale), and may include follow-up length, outcome definitions, trial quality (RoB), analysis and reporting standards (including risk of selective reporting) and the participant-level characteristics (eg, leading to trial differences in case-mix variation, including frailty). When such effect modifiers are systematically different in trials making the same comparison(s), this manifests itself as between-study heterogeneity in treatment effects. When such effect modifiers are systematically different in the subsets of trials providing direct and indirect evidence about a particular comparison, this causes inconsistency (ie, disagreement between the direct and indirect evidence for that comparison) in the NMA.

Hence, before any analysis, the distribution of potential effect modifiers will be examined across the studies to inform inconsistency concerns (whether direct and indirect evidence in the NMA are likely to be coherent) and whether some trials should be removed to improve consistency. Clearly, such decisions will also be based on the inclusion and exclusion criteria for the project.

## Data synthesis

We will meta-analyse the extracted effect estimates using modules within R and Stata, such as metafor, metan, mvmeta and network. Random-effects MAs will be conducted, to allow for potential between-study heterogeneity in each intervention effect.[26] Restricted maximum likelihood (REML) estimation will be used to fit all the models, with 95% CIs derived using an approach to account for uncertainty in the estimate of heterogeneity (tau-squared), such as the Hartung-Knapp-Sidik-Jonkman approach.[27] Initially for each outcome, we will perform a separate MA for each type of intervention, to provide summary effectiveness results based only on direct evidence. We will summarise ORs and RRs for binary outcomes, pooled (standardised) mean differences for continuous outcomes and pooled HRs for survival outcomes. We will display forest plots, with study-specific estimates, CIs and weights, alongside the summary (pooled) MA estimates, 95% CI, and (if appropriate) a 95% prediction interval.

### Network meta-analysis

An NMA will then be conducted (for each outcome separately), using a multivariate random-effects MA framework via the network module in Stata and using REML estimation (with CIs derived accounting for uncertainty of variance estimates).[28] Nodes in the network will correspond to each intervention group as outlined previously. The NMA framework allows both direct and indirect evidence to contribute toward each intervention effect (treatment contrast) via a consistency assumption.[29] The within-study correlation of multiple intervention effects from the same trial (ie, in multigroup trials) will be accounted for, and a common between-study variance assumed for all treatment contrasts in the network (thus implying a +0.5 between-study correlation for each pair of treatment effects). If possible, sensitivity to relaxing this assumption will be examined using model fit statistics. We will produce summary (pooled) effect estimates for each pair of treatments in the network, with 95% CI, and the borrowing of strength statistic to reveal the contributions of indirect evidence. For binary outcomes, if possible, we will do an NMA of both OR and RR, to check the robustness of conclusions to the choice of effect measure. Based on the results, the ranking of intervention types will be calculated using resampling methods, quantified by the probabilities of being ranked first, second, …, last, together with the mean rank and the Surface Under the

Cumulative RAnking curve, and will be presented with appropriate plots.

### Assessment of inconsistency

The consistency assumption will be examined for each treatment comparison where there is direct and indirect evidence (seen as a closed loop within the network plot). This involves estimating direct and indirect evidence, and comparing the two.[30–32] The consistency assumption will also be examined across the whole network using 'design-by-treatment interaction' models, which allow an overall significance test for inconsistency. If evidence of inconsistency is found, explanations will be sought and resolved (eg, with consideration of the distribution of effect modifiers; see earlier).

### Examination of small-study effects

If there are 10 or more studies in an MA, funnel plots will be presented to examine small-study effects (potential publication bias). Egger's, Peter's and Debray's test of asymmetry will be used for continuous, binary and survival outcomes, respectively.

### Examination of frailty impact

MA results will initially be presented for all levels combined, then for frailty/pre-frailty where reported data permit. That is, a separate MA and NMA for each frailty type will be conducted. Indeed, this may also reduce any inconsistency (see above). We will also consider extending the MA and NMA models to a meta-regression, with frailty/pre-frailty as a study-level categorical covariate allowing effects of frailty/pre-frailty to vary for each treatment effect, to quantify if intervention effects vary according to population-level frailty.

All analyses to examine frailty impact will initially be restricted to trials using a validated measure. Sensitivity analyses will (1) be restricted to trials using the phenotype model to identify pre-frailty/frailty as an internationally established reference standard, (2) include trials that used either a validated or an operationalised measure of frailty, and (3) include all trials, including by study-level categorisation of frailty status.

### Additional analyses

We will also run additional sensitivity analyses to present results of more recent evaluations, restricted to trials in the last 15 years. Meta-regression will be used to quantify differences in summary effects between studies at low RoB and other studies, and between those with shorter and longer lengths of follow-up. A multivariate NMA will be considered to accommodate all outcomes simultaneously, to examine if conclusions remain the same after accounting for the correlation among outcomes.[33] As mentioned, we will consider how relaxing the assumption of common between-study variances improves model fit.

### Confidence in cumulative evidence

We will use the GRADE framework, adapted for NMA, to rate evidence quality.[34–37] Our assessment of quality of treatment effects will enable generation of GRADE evidence profiles for our individual intervention groupings for each outcome separately.

The assessment of quality of treatment effects will include presentation and rating of the quality of direct and indirect treatment estimates separately and combined in NMA,[36] with a focus on first-order loops for assessment of indirect treatment estimates. As we will include RCTs and cRCTs, the starting point will be a high-quality evidence rating. For each estimate of treatment effect, we will assess RoB, inconsistency, indirectness, imprecision and publication bias. We will make an overall judgement on whether the quality rating for each effect warrants downgrading on the basis of limitations in each of the domains, aligned with GRADE guidance.[37] We will not consider imprecision when rating the direct and indirect estimates to inform the combined NMA rating, aligned with recent guidance.[34] Furthermore, in the presence of incoherence between direct and indirect estimates, we will assess the certainty of evidence of each estimate to guide whether or not the network estimate is downgraded.[34]

### Patient and public involvement

Our established patient and public involvement Frailty Oversight Group (FOG) provides connections to the whole spectrum of older people, with a focus on those living with frailty. We have consulted our FOG throughout the development of this protocol and discussed plans in detail at quarterly meetings. A specific example of their influence is our selection of main and additional outcomes of importance for older people. We plan to involve FOG in our intervention grouping and dissemination of this review.

### Timelines

Formal screening of search results began in January 2020. Data extraction began in May 2020. RoB assessment and data synthesis have not yet begun. We are currently updating our searches (September 2020). The project is due to complete in October 2021.

## ETHICS AND DISSEMINATION

Ethics approval is not needed as this systematic review will use aggregated, anonymised data that is available in the public domain.

This will be the first systematic review with NMA comparing the effectiveness of community-based complex interventions to sustain independence for older people, including the effect of frailty and pre-frailty. The review will use a detailed analysis to group the included interventions to identify the best configurations. Furthermore, it will also review the quality of evidence using the GRADE approach.[36] We will disseminate the findings widely through communication with healthcare providers, conference presentations and academic publications. We will adhere to PRISMA-NMA reporting guidelines.[14] Hence, this systematic review will produce

transparent and accessible results that are of great relevance and applicability for a wide audience, including policy-makers, commissioners, healthcare/social care professionals, older people and researchers working with an older population.

**Author affiliations**
[1]Academic Unit for Ageing and Stroke Research, Bradford Institute for Health Research, Bradford Teaching Hospitals NHS Foundation Trust, University of Leeds, Bradford, UK
[2]Centre for Prognosis Research, School of Medicine, Keele University, Keele, UK
[3]School of Medicine, Keele University, Keele, UK
[4]School of Medicine, University of Nottingham, Nottingham, UK
[5]Nottingham University Hospitals NHS Trust, Nottingham, UK
[6]NIHR Applied Research Collaboration East Midlands, Nottingham, UK
[7]NIHR Nottingham Biomedical Research Centre, Nottingham, UK

**Acknowledgements** We are grateful to our information specialist, Deirdre Andre, for her assistance developing the search strategy. We are also grateful to Lesley Brown and Nicola Harrison for their assistance in consulting with our Frailty Oversight Group.

**Contributors** AC and TFC were responsible for conception and design of the study and are guarantors of the review. RRiley, JE and RB designed the statistical analysis plan. NL, RB, MJ, EP, RRamiz, AF and JRFG provided critical revisions of all aspects of the review. All authors have reviewed and approved the final manuscript.

**Funding** This project is funded by the National Institute for Health Research (NIHR) Health Technology Assessment programme (project reference NIHR128862). AC is partly funded by the NIHR Applied Research Collaboration, Yorkshire and Humber, and Health Data Research UK, an initiative funded by UK Research and Innovation Councils, NIHR and the UK devolved administrations, and leading medical research charities. AF receives funding from an NIHR Senior Investigator award. The views expressed are those of the author(s) and not necessarily those of the NIHR or the Department of Health and Social Care.

**Disclaimer** The funders had no role in the design of the planned study or preparation of this protocol.

**Competing interests** None declared.

**Patient consent for publication** Not required.

**Provenance and peer review** Not commissioned; externally peer reviewed.

**ORCID iDs**
Thomas F Crocker http://orcid.org/0000-0001-7450-3143
Andrew Clegg http://orcid.org/0000-0001-5972-1097
Ram Bajpai http://orcid.org/0000-0002-1227-2703

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
