## [Reviewer comments · BMJ Open]

ARTICLE DETAILS

TITLE (PROVISIONAL)	Community-based complex interventions to sustain independence in older people, stratified by frailty: a protocol for a systematic review and network meta-analysis
AUTHORS	Crocker, Thomas; Clegg, Andrew; Riley, Richard; Lam, Natalie; Bajpai, Ram; Jordão, Magda; Patetsini, Eleftheria; Ramiz, Ridha; Ensor, Joie; Forster, Anne; Gladman, John

VERSION 1 – REVIEW

REVIEWER	Nicola Carey University of Surrey
REVIEW RETURNED	19-Nov-2020

GENERAL COMMENTS	The article presents a protocol for a systematic review and network meta analysis The paper is very well written and clearly addresses each step that inform a protocol to undertake the work. When completed the review will make an important contribution and update to evidence base in this area of practice
---

REVIEWER	Afroditi Stathi University of Birmingham, UK
REVIEW RETURNED	07-Dec-2020

GENERAL COMMENTS	The topic of this systematic review is worthy of research and reporting. Of particular strength and significance is the proposed network meta-analysis. I would suggest the authors to revisit the manuscript and provide clarifications on the following: 1. Page 3_ Keywords: Most of the keywords are the same as the title of the paper. Could you please revisit the list of keywords and provide new ones that are not part of the title?2. Page 5_ Last paragraph. Two out of the three systematic reviews cited (Refs 11-13) refer to home-based interventions. As this section refers to community-based complex interventions could you comment on the lack of more systematic reviews/individual studies focussing on out of home interventions?3. Page 7_ Intervention. Could you please explain in more detail what is classified under the terms: Intervention practices, structural elements and contextual factors? As it stands at the moment, this eligibility criterion is quite abstract.4. Page 8_ Bullet point 2. Would an exercise intervention with a behavioural component be eligible or not? Can you further explain/refine this exclusion criterion?
---

	5. Page 8_Outcomes: I was surprised not to see objective measure of physical function not included as one of the outcomes as frailty is the focus of this work. Could you justify this please? 6. Page 10_Assessment of frailty: Could you add a more detailed list of expected validated measures based on previous systematic reviews and knowledge of recently published literature in the topic of frailty?
--	---

VERSION 1 – AUTHOR RESPONSE

Nicola Carey review:

We thank the reviewer for their consideration and kind comments.

Afroditi Stathi review:

1. Page 3_Keywords: Most of the keywords are the same as the title of the paper. Could you please revisit the list of keywords and provide new ones that are not part of the title?	We thank the reviewer for their very thoughtful suggestions. We have updated the list of keywords (page 3) with terms that will hopefully make this manuscript more accessible from a broader range of searches, should it be published.
2. Page 5_Last paragraph. Two out of the three systematic reviews cited (Refs 11-13) refer to home based interventions. As this section refers to community based complex interventions could you comment on the lack of more systematic reviews/individual studies focussing on out of home interventions?	Thank you. On page 5 we have added: “Moreover, few systematic reviews have investigated such interventions delivered outside of the home environment.” Before the “A landmark 2008...” sentence.
3. Page 7_Intervention. Could you please explain in more detail what is classified under the terms: Intervention practices, structural elements and contextual factors? As it stands at the moment, this eligibility criterion is quite abstract.	Thank you. We have added details in the paragraph below (pages 7-8) about these terms and provided some examples to help illustrate how this might be operationalised. See also our answer to query 4 below.

4. Page 8_Bullet point 2. Would an exercise intervention with a behavioural component be eligible or not? Can you further explain/refine this exclusion criterion?	We distinguish aspects of components from discrete components and have added the term discrete to this bullet point. For illustration, most exercise interventions run as group sessions involve multiple behavioural techniques such as instruction on how to perform the behaviour, demonstration of the behaviour, graded tasks; we do not consider these different components as they occur as an integral part of the practice. Should there have been such group sessions and a separate goal setting process not solely relating to the sessions we would consider the intervention eligible in this regard. For instance, The REirement in ACTion (REACT) intervention (Stathi et al. Trials 19:228) is eligible because it has both physical activity sessions and educational/social sessions. We hope this sufficiently explains the criterion.
5. Page 8_Outcomes: I was surprised not to see objective measure of physical function not included as one of the outcomes as frailty is the focus of this work. Could you justify this please?	While frailty is a focus of this work, that is in relation to the population; we have not used frailty as an outcome. In our consultations with PPI members they did not mention physical function in this sense. While such measures can be useful predictors or indicators of patient-important outcomes we do not think change in an objective measure of physical function is as important to stakeholders as change in the outcomes we have selected. We recognise that there can be no one right perspective on this but seek to not list too many outcomes, particularly given the multiple timepoints we have proposed.
6. Page 10_Assessment of frailty: Could you add a more detailed list of expected validated measures based on previous systematic reviews and knowledge of recently published literature in the topic of frailty?	Thank you very much. Your query has prompted us to add detail to our approach , as well as providing a working definition of validated measures and a list to exemplify this based on a recent systematic review. We also noticed that gait speed had slipped into the wrong list so are grateful to have been given the opportunity to correct this (pages 10-11). We thank the reviewer again for their detailed and careful consideration of this manuscript.

Additional changes:

We have made an addition to the funding statement and slightly reorganised this text. (p17)